# The Effects of High-Fat Diets from Calcium Salts of Palm Oil on Milk Yields, Rumen Environment, and Digestibility of High-Yielding Dairy Cows Fed Low-Forage Diet

**DOI:** 10.3390/ani12162081

**Published:** 2022-08-15

**Authors:** Eyal Frank, Lilya Livshitz, Yuri Portnick, Hadar Kamer, Tamir Alon, Uzi Moallem

**Affiliations:** 1Department of Ruminant Science, Institute of Animal Sciences, The Volcani Institute, 68 HaMaccabim Road, P.O. Box 15159, Rishon LeZion 7505101, Israel; 2Department of Animal Science, The Robert H. Smith Faculty of Agriculture, Food and Environment, The Hebrew University of Jerusalem, Rehovot 76100, Israel

**Keywords:** fat supplementation, high-fat diet, calcium salt of fatty acids, rumen fermentation, nutrient digestibility

## Abstract

**Simple Summary:**

Calcium salts of fatty acids from palm oil represent a very common supplement in dairy cows’ diets. The present results show that including up to 3.9% of CS-PFA in lactating cow rations with low fiber content did not affect milk and fat yields, decreased protein percentages and yields, reduced rumen fermentation, and decreased the total-tract apparent digestibility of NDF. In spite of the negative effects on rumen fermentation and digestibility, the adverse effects on yields were minor, which indicates that, under specific circumstances, feeding high-fat diets containing large amounts of CS-PFA to dairy cows fed low-fiber diets is possible.

**Abstract:**

Instability in grain prices led to continuing worldwide growth in the proportion of fat supplements in lactating cows’ rations. However, fat supplementation was associated with decreases in feed intake, rumen fermentation, and feed digestibility. The present objectives were to test the effects of high-fat diets from calcium salts of palm oil fatty acids (**CS-PFA**) in lactating cow rations containing high proportions of concentrate, on feed intake, milk yields, rumen environment, and digestibility. Forty-two multiparous mid-lactation dairy cows were assigned to three treatments, designated as low fat (**LF**), moderate fat (**MF**), and high fat (**HF**) that contained (on DM basis), respectively, (i) 4.7% total fat with 1.7% CS-PFA, (ii) 5.8% total fat with 2.8% CS-PFA, and (iii) 6.8% total fat with 3.9% CS-PFA. Rumen samples were collected for pH, ammonia, and volatile fatty acid (**VFA**) measurements, and fecal grab samples were collected for digestibility measurements. A numerical trend of decreasing dry matter intake with increasing CS-PFA in diet was observed: 28.7, 28.5, and 28.1 kg/day in LF, MF, and HF, respectively (*p* < 0.20). No differences between treatments were observed in milk yields and milk-fat percentages, but protein percentage in milk tended to fall with increasing dietary CS-PFA content (*p* < 0.08), which resulted in 6.4% smaller protein yields in the HF than in the LF group (*p* < 0.01). Milk urea nitrogen was 15.3% higher in HF than in LF cows (*p* < 0.05). Rumen pH was higher at all sampling times in the MF and HF than in the LF cows. Concentrations of propionic acid and total VFA were higher in LF than in MF and HF cows. The apparent total-tract digestibility of dry matter was higher with LF than with HF (*p* < 0.002), and that of organic matter was lowest with the HF diet (*p* < 0.005). The apparent NDF digestibility declined with increasing dietary fat content, and it was 8.5 percentage points lower in HF than in LF cows (*p* < 0.009). Apparent fat digestibility increased with increasing dietary fat content, and it was higher by 10.4 percentage points in the HF than in the LF group (*p* < 0.004). In conclusion, diets with high concentrate-to-forage ratios, containing up to 6.8% total fat and 3.9% CS-PFA, negatively affected rumen fermentation and NDF digestibility in high-yielding dairy cows; however, the effects on yields were minor, indicating that, under specific circumstances, the inclusion of large amounts of CS-PFA in dairy cows’ rations with low fiber content is feasible.

## 1. Introduction

Fat supplements are regarded as energy sources in ruminants’ diets and form a very common component of lactating cows’ rations. The increasing energy requirements of the high-yielding dairy cow and fluctuations in grain prices led to continuing growth in the use of such fat supplements. Calcium salts of palm oil fatty acids (**CS-PFA**) are the most commonly used fat supplement in dairy cows’ rations in Israel. Calcium salts of long-chain fatty acids (FA) comprise FA complexed with a calcium ion, which makes them insoluble. Microbes cannot absorb FA as calcium salts and FA salts have little effect on microbial fermentation [1].

Although fat supplements differ markedly in their effects on dry matter intake (DMI), all fat supplements, even those considered to be rumen-inert, generally tend to reduce DMI [2,3,4,5]. Schauff and Clark [3] found that DMI of dairy cows decreased linearly with increasing supplementation with calcium salts of long-chain fatty acids. Furthermore, in several studies, fat supplementation depressed rumen fermentation and digestibility of diet components [2,3,6]. The typical Israeli dairy cow ration includes high concentrates and low forage contents: 64–67% and 33–36%, respectively. Fat supplementation is more detrimental to fiber than to that of nonstructural carbohydrate diet components [6], which raises the possibility of reducing adverse effects of fat supplementation on digestibility in diets containing low proportions of fiber. Therefore, we hypothesized that increasing the CS-PFA to a very high level in low-forage diets would not have any detrimental effects on cows’ performance or health. Therefore, the objectives of the present study were to test the effects of feeding lactating cows with high-fat diets with high concentrate-to-forage ratios (~68:32, respectively) and containing high proportions of CS-PFA, on feed intake, milk yields, rumen environment, and apparent total-tract digestibility of nutrients.

## 2. Materials and Methods

### 2.1. Animals and Experimental Procedures

The protocol of the experiment was approved by the Volcani Center Animal Care Committee (IL 0358/14), and the study was conducted at the Volcani Center experimental farm in Bet-Dagan, Israel. Forty-two multiparous Israeli-Holstein cows that averaged 131 ± 55 days in milk (DIM) were group-housed in shaded loose pens with adjacent outside yards, which were equipped with a real-time electronic individual feeding system. Each feeding station was equipped with an individual identification system (I.D tag; S.A.E., Kibutz Afikim, Israel) that enabled each cow to enter a specific feeding station and automatically recorded each meal. After adaptation to the feeding system, the cows were fed the same diet for 10 days as a covariate period; afterward, they were assigned to three treatment groups, blocked by milk yield, DIM, parity, and BW. The cows were fed standard Israeli dairy cow rations as TMR, designated as (i) low fat (LF), (ii) medium fat (MF), and (iii) high fat (HF). These diets had similar contents, but they contained differing fat proportions and amounts of CS-PFA (on DM basis of the total diet), as follows: (i) 4.7% fat with 1.7% CS-PFA; (ii) 5.8% fat with 2.8% CS-PFA; (iii) 6.8% fat with 3.9% CS-PFA. The CS-PFA was produced by Poliva Ltd. (Ramla, Israel). The fatty-acid profile of the CS-PFA supplement was 1% myristic (C14:0), 45% palmitic (C16:0), 18% stearic (C18:0), 40% oleic (C18:1), and 9% linoleic (C18:2). The experimental diet composition and content are presented in Table 1; the diets were isonitrogenous, and energy content was slightly higher in the HF diet than the others. The NE_L_ values of NRC (1989) [7] were used for diet formulation.

### 2.2. Measurements

The diets were supplied as a total mixed ration (TMR) once daily at 10:00 a.m. for 12 weeks. They were supplied at 106% of the expected intake, which was adjusted according to the previous day’s intake. Daily individual intake was recorded according to offered and leftover amounts, and feed efficiency was determined for each cow by calculating the milk or 4% fat-corrected milk (FCM) yield per kilogram of dry matter (DM) consumed. The cows were milked three times daily, milk yields were recorded electronically, and the cows were weighed automatically after each milking, using a walk-in electronic scale (S.A.E. Afikim, Kibbutz Afikim, Israel). The cows were equipped with a Plus sensor (S.A.E. Afikim, Kibbutz Afikim, Israel) that monitored and recorded their lying time (resting time) as min/day. Milk samples were collected from three consecutive milkings every 2 weeks and were analyzed for milk fat, protein, lactose, and urea contents by infrared analysis (standard IDF 141C:2000) at the laboratories of the Israeli Cattle Breeders’ Association (Caesarea, Israel). Somatic cell counts were determined in the same laboratory.

### 2.3. Energy Calculations

Energy content in milk, and energy balance (**EB**) were calculated according to the NRC (2001) [8] as follows:NE_c_ = NE_L_ per kg DM × DMI,
NE_m_ = BW^0.75^ × 0.08 × 1.1,
NE_p_ = milk (kg) × [{0.0929 × (fat%)} + {0.0547 × (protein%)} + {0.0395 × (lactose%)}],
EB = NE_c_ − (NE_m_ + NE_p_),
where NE_c_ is the net energy consumed, NE_m_ is the net energy required for maintenance, and NE_p_ is the net energy output in milk.

### 2.4. Rumen Sampling

During the 12th week of the study, trained personnel used vacuum stomach pumps to collect 400 to 600 mL samples of rumen fluid from each of 30 cows, i.e., 10 animals from each treatment group. The samples were collected on the same day at 10:00 a.m., 2:00 p.m., and 4:00 p.m., i.e., 2 h before, and 2 and 4 h after feeding. Precautions were taken to avoid contamination of rumen samples with saliva; the pump was turned on only after ensuring that the tube was positioned inside the rumen, and the rumen samples were tested visually for color and texture. If there was any suspicion of saliva contamination, the sample was discarded and a new rumen sample was collected. Ruminal pH was determined immediately after sample collection using an EL20 pH meter (Mettler-Toledo Instruments, Shanghai, China). For ammonia measurements, 5 mL of filtered rumen fluid was mixed immediately after collection with an equal volume of 20% trichloroacetic acid (TCA), and stored at −32 °C pending analysis. A second filtered ruminal fluid sample was preserved with HgCl_2_, centrifuged for 10 min at 1000× *g*, and stored at −32 °C pending analysis for volatile fatty acids (VFA).

### 2.5. Total-Tract Digestion Fecal Samples Collection and Analysis

During the ninth week of the experiment, three fecal grab samples were collected from each of 10 cows in each treatment group, at 3 h intervals during each of three consecutive days, i.e., nine samples from each cow. The fecal samples were dried at 60 °C for 48 h in a forced-air oven, and then ground to pass through a 1.0 mm S-M-100 sieve (Retsch, Haan, Germany). Simultaneously, diet samples were collected during the 3 days of fecal sampling, dried, and ground. For in situ measurements, 0.5 g of each dried, ground fecal sample from each cow, i.e., nine samples, totaling 4.5 g, were combined and placed in a 6 × 12 cm Dacron bag that had 42–44 µm pores (Emka, Petah Tikva, Israel), for incubation in triplicate. Three additional Dacron bags were each filled with a 4.5 g sample from each diet for in situ analysis, i.e., analysis in triplicate. The bags were incubated together in a rumen-cannulated cow and removed after 192 h. They were then washed in a washing machine, dried at 60 °C for 48 h, and weighed. The indigestible neutral detergent fiber (I-NDF) was used as a marker for apparent total-tract digestibility analysis, and the residuals were analyzed for I-NDF [9,10,11]. The fecal and ration samples were analyzed for DM, protein, NDF, ADF, ether extract, and ash content. The digested amounts of each chemical component of the ration were calculated individually using the average individual DMI, as determined during the 3 days of collecting fecal grab samples.

### 2.6. Chemical Analysis 

Total mixed rations were sampled weekly for determination of DM, CP, NDF, ADF, calcium (Ca), and phosphorus (P). The feed samples were dried at 65 °C for 24 h, and then ground to pass through a 1.0 mm S-M-100 sieve (Retsch). The ground samples were dried at 100 °C for 24 h and analyzed for the various contents as follows: N according to the Association of Official Analytical Chemists (AOAC) [12] method 984.13; NDF and ADF contents using Ankom equipment (Ankom Technology, Fairport, NY, USA); NDF using α-amylase and sodium sulfite according to Van Soest et al. [13]; Ca according to the AOAC [12], method 935.13; P according to the AOAC [12], method 964.06; ether extracts according to the AOAC [12], method 996.06. Samples were dried at 550 °C for 3 h for ash determination. The NE_L_ (net energy for lactation) value of feedstuffs was determined according to the National Research Council (NRC) (1989) [7], as is usual in Israel. Volatile fatty acids in centrifuged ruminal fluid were assessed using a 5890 series 2 gas chromatograph (Agilent Technologies, Wilmington, DE, USA) equipped with a capillary column (30 m × 0.53 mm, 0.5 mm i.d.; Agilent Technologies, Santa Clara, CA, USA) and a flame ionization detector (FID); the injection port, column, and detector were maintained at 175, 130, and 165 °C, respectively. Rumen ammonia concentrations were determined using the phenol procedure [14].

### 2.7. Statistical Analyses

Continuous variables (milk, milk solids, DMI, and efficiency variables) were analyzed as repeated measurements with Proc Mixed software, version 9.2 (SAS, 2002) [15]. When relevant, variables were analyzed with the specific data of the pretreatment period as covariates.

The model used was as follows:Y_ijkl_ = µ + T_i_ + L_j_ + C(T × L)_ijk_ + DIM_ijkl_ + *E*_ijklm_,
where µ is the overall mean, T_i_ is the treatment effect (i = 1–3), L_j_ is the parity (j = 2 or >2), C(T × L)_ijk_ is cow k nested in treatment i and cow k nested in parity j, DIM_ijkl_ is the day in milk as a continuous variable, and *E*_ijklm_ is the random residual.

The interactions (treatment × parity) and (treatment × DIM) were tested, found not significant; therefore, they were excluded from the model.

Rumen measurements (pH, and ammonia and VFA concentrations) were analyzed as repeated measurements with Proc Mixed software, version 9.2 (SAS, 2002) [15]. Nutrient digestibility was analyzed using the general linear model (GLM) procedure software version 9.2 (SAS, 2002) [15]. Least squares means and adjusted SEM are presented in Table 2, Table 3, Table 4 and Table 5; *p* < 0.05 was accepted as significant unless otherwise stated.

## 3. Results and Discussion

Feeding high-yielding dairy cows diets with high concentrate-to-forage ratios, containing up to 6.8% total fat and 3.9% CS-PFA, did not impact milk and fat yields (Table 2), but reduced milk protein yields, and increased the MUN content. In the rumen, the pH increased and total VFA concentration decreased with increasing dietary fat content. In addition, total-tract digestibility of DM, OM, and NDF decreased as intake of CS-PFA increased, but fat digestibility was enhanced.

### 3.1. Effects on Dry Matter Intake

No significant between-group differences in DMI or estimated energy intake were observed. There are substantial differences between fat products [16]; therefore, the present discussion is focused mainly on the effects of supplementation with calcium salts of fatty acids, and especially CS-PFA. Chouinard et al. [17] examined the effect of increasing the percentages of calcium salts of canola oil in diets from 0% to 4% and found that DMI decreased linearly as the fat percentage increased, similarly to the trend found by Schauff and Clark [3], who used increasing amounts of calcium salts of long-chain fatty acids. Hammon et al. [18] found that cows fed a diet containing 7.2% total fat consumed 4.7% less DM than those receiving 2.4% total fat in their diet. Harvatine and Allen [19] compared a diet containing 6.8% total fat that included 4% Ca-PFA with one containing prilled fat, i.e., 7.1% total fat, and they found 2.6% less DMI in the cows receiving the former diet. Taken together, these findings are in accordance with the meta-analysis results of Rabiee et al. [16], who found that DMI diminished as a result of feeding various fat products, including CS-PFA. Allen [5] estimated that DMI is depressed by approximately 2.5% compared with controls, for each percentage point of CS-PFA added to the diet. The reasons why supplemental fat, especially CS-PFA, reduced feed intake are not completely clear; however, CS-PFA emits a characteristic and exceptional odor that may influence the acceptability of this supplement [20]. Furthermore, in the present study, the fat supplement was included in the TMR; as also suggested by Allen [5], including the supplemental fat in the TMR might reduce the negative impact on acceptability. Several other reasons suggested for the hypophagic effect of supplemental fat include fat composition and the adverse effect on digestibility.

### 3.2. Effects on Yields of Milk and Milk Solids

In the present study, no between-group differences in milk yields and milk fat percentages associated with feeding high rates of CS-PFA were observed (Table 2). Several previous studies of the effects of supplementation of various fat products at high rates on dairy cows’ performance obtained inconsistent results. Sklan et al. [21] found that feeding an additional 2.5% of CS-PFA (making 4.9% total fat in the diet) to multiparous cows increased milk yields by 9.6% and fat yields by 15%, and tended to increase milk fat and decrease milk protein percentages. In another study, Wu et al. [22] reported that supplementation of 2.4% CS-PFA (making 6.1% total fat in diet) increased milk and fat yields by 4% and 8.8%, respectively, and decreased protein percentage, without affecting fat percentage or protein yields. Schneider et al. [23] compared cows supplemented with Ca-PFA at 0.5 or 0.45 kg/day per cow with control animals fed a diet containing 4.6% fat, and found 4.4% to 4.9% increases in milk yields and 6% to 9% increases in 3.5% FCM yields. Lohrenz et al. [24] fed mid-lactation cows with a diet containing 5.7% added CS-PFA that provided 6% total fat in the diet, compared with 2.2% in controls; they observed no differences in milk yields or in fat percentages and yields, but found lower protein percentages in the CS-PFA cows’ milk. Similarly to Lohrenz et al. [24], in the present study, we observed no impact on milk and fat yields as a result of inclusion of 2.8% or 3.9% CS-PFA in diets. The meta-analysis of Rabiee et al. [16] found that feeding CS-PFA elicited an actual milk yield increase of 0.55 kg/day per cow. The inconsistencies among reported changes in milk yields and in fat percentages and yields that resulted from feeding CS-PFA may be related to variability in study design regarding total fat percentage in diet and the effects on intake. It should be noted that, in several studies, the fat supplement was provided as top-dressing, and the energy density of the fat-supplemented diets was higher than that of the controls, which had confounding effects on energy intake and, consequently, on yields. In the present study, similarly to the findings of Wu et al. [22] and Lohrenz et al. [24], we did not observe any impact of feeding high rates of CS-PFA on fat percentages or yields. Hammon et al. [18] reported a lower milk-fat percentage from cows fed a diet containing 7.2% total fat than from those fed one containing 2.4%, whereas Schneider et al. [23] found an increase in milk fat associated with feeding CS-PFA. These variations in milk-fat results may be related to differing effects on milk yields and differing energy partitioning between production of milk and milk fat, respectively.

In the present study, a trend toward reduced protein percentage in milk was found with increasing dietary fat content (3.25%, 3.16%, and 3.13% in LF, MF, and HF groups, respectively; *p* < 0.08), which resulted in 6.4% lower protein yields in the HF than in the LF group (*p* < 0.01; Table 2). Wu and Huber [25] reported that the depression of milk protein content occurred regardless of the type of fat fed, whereas Rabiee et al. [16] reported that the largest negative effect on protein percentage was elicited by calcium salts of fatty acids. The relationship between dietary fat supplementation and milk protein concentration, as well as possible mechanisms, was reviewed thoroughly by Wu and Huber [25], who suggested mechanisms involving rumen and mammary gland metabolism, and concluded that insufficient availability of amino acids might be the major reason for the decrease in milk protein. In the present study, the casein percentage in milk was higher in the LH than in the HF cows (*p* < 0.05), and similar to that in the MF group (2.49%, 2.33%, and 2.39%, respectively; SEM = 0.04). Casein is synthesized de novo in the mammary gland, and it contributes 77% to 81% of total milk N. This trend was also reported by Dunkley et al. [26], DePeters et al. [27], and Chow et al. [28], which suggests that the depression in milk protein content that resulted from high-fat diets occurred in the mammary gland. In the present study, the milk urea nitrogen (MUN) concentrations increased with increasing dietary fat content (*p* < 0.05) and were 15.3% higher in the HF than in the LF group (*p* < 0.01). A positive relationship between MUN concentrations and excretion of N in urine was found by Ciszuk and Gebregziabher [29], which means that, in the present study, the urine N excretion was, most likely, higher in the HF than in the LF cows. This occurred although the rumen ammonia concentrations did not differ between groups (Table 4), which suggests that higher N excretion apparently occurred although no more ammonia was available for urea synthesis in the HF cow liver than in the LF one. The present findings support the premise that the availability of N sources, i.e., amino acids, was lower in high-fat than in low-fat diets, which would be consistent with the higher N excretion in the milk and, most likely, urine of the HF cows.

### 3.3. Effects on Efficiency

Efficiency calculations are summarized in Table 3. Conversion of DMI to milk did not differ between groups, but conversion of DMI or energy consumed to 4% FCM was higher in MF than in LF (*p* < 0.01). Weiss et al. [30] reported that supplementation of calcium salts of fatty acids (with about 6% total fat in the diet) increased efficiency of conversion of DMI into milk or FCM (4%), similarly to previous findings of Moallem et al. [31], who used a diet supplemented with CS-PFA that provided 6.6% of the total fat in the diet compared with 5.1% in controls. Furthermore, Wu et al. [22] found that feeding high rates of CS-PFA increased the efficiency of conversion of DMI into FCM (but not into milk). However, others did not find that feeding high-fat diets increased conversion efficiency [32,33]. These differences among various workers’ results may be attributed to differences between studies, in stages of lactation and levels of production.

The total BW gain through the whole of the present study tended to be higher in LF than in HF cows (27.3 vs. 16.8 kg; *p* < 0.1; Table 3), which is surprising because the calculated energy balances were similar between groups. Nevertheless, this trend of lower BW gain resulting from diets in which high fat content replaced carbohydrates was found elsewhere [31,34]; this indicates that there was differing partitioning of energy resources, i.e., less energy allocated to body mass retention under high-fat diets.

### 3.4. Rumen Environment

Rumen pH and concentrations of ammonia and VFA are presented in Table 4; values of rumen pH and concentrations of propionate and VFA, as they varied with sampling time, are presented in Figure 1 (Panels, A, B, and C, respectively). Rumen pH was higher in MF and HF than in LF cows at all sampling times. Very few previous studies examined the effect of high-fat diets on rumen pH; Ohajuruka et al. [35] observed no differences in rumen pH between cows fed additional 3.40% or 5.92% CS-PFA, which provided total dietary fat of 5.25% or 7.29%, respectively, whereas Elliott et al. [32] found a tendency for pH to fall as fat supplementation increased. Rico et al. [11] also found that no changes in rumen pH resulted from feeding CS-PFA. In the present study, CS-PFA replaced concentrates as a source of energy, and the percentage of grain decreased from 27.7% in the LF to 21.4% in the HF diet. The lower grain content in the high-fat diets decreased the fermentation rate of starch in the rumen, as can be seen in Figure 1. Fat supplementation reduced rumen VFA production, and, at 2 and 4 h after feeding, propionate and total VFA concentrations were much lower in MF and HF than in LF cows (Figure 1), which helped to sustain the high rumen pH in the cows fed high-fat diets. The average concentration of acetic acid was higher in the LF than in the MF group (*p* < 0.04) and tended to be higher than in the HF (*p* < 0.1). The concentrations of propionic acid and total VFA were higher in LF than in MF and HF cows (*p* < 0.05). Harvatine and Allen [33] also observed lower total VFA in cows fed high-fat diets, and Schauff and Clark [3] found lower propionate concentrations in the rumens of cows fed Ca-LCFA; however, Elliott et al. [32] found only minor effects of fat supplementation on ruminal fermentation. As described above, the shift from high concentrate in the LF diet to low concentrate in the HF one influenced the rate and turnover of ruminal fermentation, resulting in decreased concentrations of individual and total VFA in the high-fat diets. Jenkins [6] suggested that the lower rumen fermentation associated with high-fat diets either resulted from coating of feed particles or was a direct toxic effect on ruminal microorganisms.

### 3.5. Effects on Apparent Total-Tract Digestibility

The apparent total-tract digestibility of nutrients and total digesta intake are presented in Table 5. The apparent digestibility of DM in the LF diet was higher than that in the HF (*p* < 0.002) and tended to be higher than that in the MF diet (*p* < 0.09). The apparent digestibility of organic matter (**OM**) was similar in the LF and MF diets, and higher than that in the HF, whereas no between-treatment differences in protein digestibility were observed. Apparent digestibility of NDF declined with increasing dietary fat content and was 8.5 percentage points lower with the HF than with the LF diet (*p* < 0.001). The apparent digestibility of fat increased with increasing dietary fat content and was as much as 10.4 percentage points higher in the HF than in the LF diet. The apparent digestible DM intake was higher with the LF than with the HF diet (*p* < 0.007), and the digestible OM intake tended to be lower in the HF than in the MF (*p* < 0.1) and LF diets (*p* < 0.07). The digestible intake of NDF tended to be higher in MF than in HF (*p* < 0.09), with no other between-group difference. Moreover, the digestible fat intake was, as expected, highest in the HF, intermediate in the MF, and lowest in the LF group (*p* < 0.006).

The digestibility of components of diets that included fat was investigated by several researchers. Harvatine et al. [33] compared a diet containing 5.5% total FA with ones containing about 8% total FA, contributed by calcium salts of fatty acids (CSFA) that had various degrees of saturation, and they found no differences in digestibility of DM, OM, or NDF; however, observed differences in total FA digestibility were related to the source of fat rather than to its proportion in the diet. Rico et al. [11] compared a diet containing 3.8% total FA with one containing 5.7% (contributed by an additional 2.3% of CS-PFA) as fed to high-yielding cows, and found no differences in digestibility of DM or NDF. Furthermore, Wu et al. [22] observed no effect of supplemental fat on DM, NDF, or ADF digestibility. Weiss et al. [30] found that supplementation of 3.6% CSFA, for a total fat content of 6.23%, compared with 2.94% fat in the control, did not affect digestibility of DM, OM, NDF, or protein; however, fat digestibility increased with increasing dietary fat content, similarly to the present findings. Ohajuruka et al. [35] also observed no effects on digestibility of DM, OM, or protein as a result of feeding CSFA. Jenkins and Palmquist [36] found no depression of digestibility of DM or ADF when CSFA was fed to dairy cows; however, Jenkins [6] concluded that general lipids added to ruminant diets could reduce digestibility of diet components, especially structural carbohydrates. In our present study, feeding dairy cows that consumed an average >28 kg/day with high rates of CS-PFA decreased apparent total-tract digestibility of DM, OM, and NDF. This digestibility was less in the HF than in the LF cows, by 9.4, 9.6, and 18.8 percentage points for DM, OM, and NDF, respectively. This relatively dramatic depression in digestibility was not observed in many previous studies; it partly could be attributed to the decrease in rumen fermentation that was manifested in lower specific and total VFA concentrations, which were lower in the HF than in the LF group of cows, the latter being 5.5% lower in the HF than in the LF (*p* < 0.05). No between-group difference in protein digestibility was observed, and fat digestibility was enhanced by fat supplementation, which indicates that the depression in DM and OM digestibility was predominantly caused by the depression in NDF digestibility. The depression in fiber digestibility that resulted from fat supplementation in ruminants is typical of natural fats rather than rumen-inert fats. Moreover, the form of calcium salts was characterized as a weak suppressor of digestibility in several publications [2,6]. However, it is well agreed that a portion, even of inert fats, becomes dissolved in the rumen and thereby exposed to biohydrogenation by ruminal bacteria [19,37,38]. Thus, it might be that, with very-high-fat diets that led to high fat consumption, as in the present study, where the average intake of CS-PFA was up to 1095 kg/day per cow in the HF group, the absolute amount of fat dissolved in the rumen was so large that it might have interfered with fiber digestibility. Furthermore, the adverse effects on fermentation and digestibility found in the present study may be attributed to the product quality, which should be tested in a comparative study alongside other similar products.

## 4. Conclusions

In conclusion, feeding mid-lactation dairy cows diets with high concentrate-to-forage ratios, containing up to 6.8% total fat and 3.9% CS-PFA, did not affect milk and fat yields, but decreased protein percentages and yields, reduced rumen fermentation, and greatly decreased the total-tract apparent digestibility of NDF. Although the cows receiving the HF diet consumed on average 1095 g of CS-PFA daily and showed negative effects on digestibility of structural carbohydrates, the adverse effects on yields were minor, which suggests that, under specific circumstances, inclusion of high proportions of CS-PFA in dairy cow diets with low fiber content would be a possible option.

## Figures and Tables

**Figure 1 animals-12-02081-f001:**
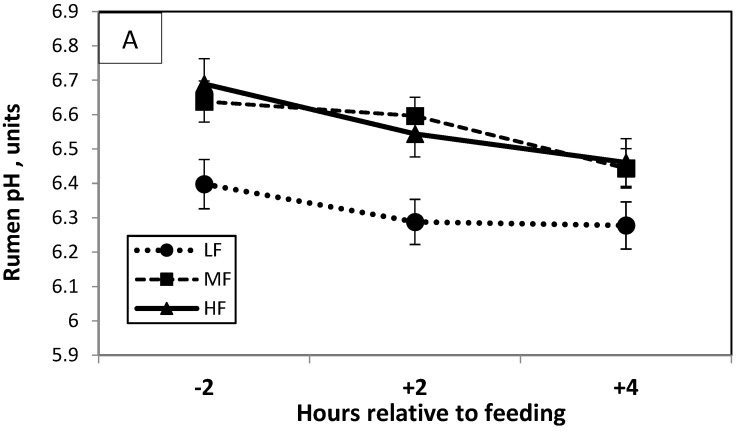
Rumen pH (**A**) and concentrations of propionate (**B**) and total volatile fatty acids (VFA) (**C**) in rumen, plotted against sampling time (−2, +2, and +4 h relative to feeding) in dairy cows fed a lactating cow ration containing (i) low fat (LF)—1.7% calcium salts of palm oil fatty acids (CS-PFA) (4.7% total fat in diet), (ii) moderate fat (MF)—2.8% CS-PFA (5.8% total fat in diet), or (iii) high fat (HF)—3.9% CS-PFA (6.8% total fat in diet).

**Table 1 animals-12-02081-t001:** Ingredients and chemical compositions of the experimental diets.

Ingredients	Treatments ^1^		
	LF	MF	HF
% of DM			
Corn, ground	21.1	23.2	19.2
Barley, rolled	3.3	1.3	1.1
Wheat grain, rolled	3.3	1.3	1.1
Rapeseed	2.2	2.2	1.9
Soybean meal	1.8	0.9	0.7
Sunflower meal	2.0	6.1	6.9
Wheat bran	0	7.8	12.5
Gluten feed	14.2	5.4	4.4
Cottonseed	2.1	2.9	2.4
Wheat silage	7.9	7.9	6.5
Corn silage	8.4	8.2	6.8
Oat hay	12.5	12.5	10.3
Clover hay	3.3	3.3	2.8
Wheat straw	0	0	3.9
DDG	9.7	10.1	12.3
By product of dairy industry	3.9	1.1	0.9
CS-PFA	1.7	2.8	3.9
Urea	0.4	0.4	0.3
Limestone	0.7	1.0	0.9
NaCl	0.6	0.3	0.3
Bicarbonate	0.8	0.8	0.8
Vitamins and minerals ^2^	0.9	0.9	0.9
Chemical composition			
NE_L_ (Mcal/kg dry matter)	1.78	1.78	1.80
Crude protein, %	16.5	16.5	16.5
Forage, %	32.1	31.9	30.3
NDF, %	30.2	32.6	34.8
Forage NDF, %	17.3	17.3	17.3
Ether extract, %	4.7	5.8	6.8
Ca, %	0.009	0.01	0.01
P, %	0.004	0.005	0.006

^1^ Treatments: Dairy cows were fed a lactating cow ration containing (i) low fat (LF)—1.7% calcium salts of palm oil fatty acids (CS-PFA) (4.7% total fat in diet), (ii) moderate fat (MF)—2.8% CS-PFA (5.8% total fat in diet), or (iii) high fat (HF)—3.9% CS-PFA (6.8% total fat in diet). DM = dry matter; DDG = dried distiller’s grains; CS-PFA = calcium salts of palm fatty acids; MJ = megajoule. ^2^ Contained: vitamins (IU/kg): A—20,000,000; D—2,000,000; E—15,000; minerals (mg/kg): Mn—6000; Zn—6000; Fe—2000; Cu—1500; I—120; Se—50; Co—20.

**Table 2 animals-12-02081-t002:** Least squares means of milk and milk solid contents and yields.

	Treatments ^1^				
	LF	MF	HF	SEM	*p*<
Milk (kg/day)	41.1	41.0	40.6	0.34	0.63
Fat (%)	3.66	3.67	3.76	0.6	0.48
Protein (%)	3.25	3.16	3.13	0.4	0.08
Lactose (%)	4.98	4.97	4.92	0.3	0.38
FCM 4% (kg/day)	38.3	39.1	38.6	0.41	0.38
Fat (kg/day)	1.52	1.50	1.51	0.02	0.90
Protein (kg/day)	1.33 ^a^	1.29 ^a^	1.25 ^b^	0.01	0.01
Lactose (kg/day)	2.05	2.03	1.99	0.02	0.24
MUN (mg/dL)	13.7 ^b^	14.1 ^b^	15.8 ^a^	0.70	0.05

Different superscript lowercase letters within a row indicate significant differences according to the *p*-value in the last column. ^1^ Treatments: Dairy cows were fed a lactating cow ration containing (i) low fat (LF)—1.7% calcium salts of palm oil fatty acids (CS-PFA) (4.7% total fat in diet), (ii) moderate fat (MF)—2.8% CS-PFA (5.8% total fat in diet), or (iii) high fat (HF)—3.9% CS-PFA (6.8% total fat in diet). FCM = fat-corrected milk; MUN = milk urea nitrogen.

**Table 3 animals-12-02081-t003:** Least squares means of dry matter intake, energy balance, and efficiency calculations.

	Treatments ^1^				
	LF ^1^	MF	HF	SEM	*p*<
DMI (kg/day)	28.7	28.5	28.1	0.22	0.20
Energy intake (Mcal/day)	50.7	50.4	51.1	1.5	0.48
EB (MJ/day)	11.4	11.3	11.5	0.36	0.91
BW gain (kg)	27.3	19.4	16.8	4.8	0.13
Milk/DMI (kg/kg)	1.45	1.46	1.46	0.01	0.84
FCM^4^/DMI (kg/kg)	1.35 ^b^	1.40 ^a^	1.39 ^ab^	0.01	0.01
FCM/Energy intake (kg/MJ)	0.182 ^b^	0.189 ^a^	0.184 ^ab^	0.001	0.03
ECM/energy intake, (MJ/MJ)	0.57	0.58	0.57	0.005	0.25
Lying time (min/day)	622.3 ^a^	577.7 ^b^	591.2 ^b^	8.0	0.002

Different superscript lowercase letters within a row indicate significant difference according to the *p*-value in the last column. ^1^ Treatments: Dairy cows were fed a lactating cow ration containing (i) low fat (LF)—1.7% calcium salts of palm oil fatty acids (CS-PFA) (4.7% total fat in diet), (ii) moderate fat (MF)—2.8% CS-PFA (5.8% total fat in diet), or (iii) high fat (HF)—3.9% CS-PFA (6.8% total fat in diet). FCM = fat corrected milk; MUN = milk urea nitrogen.

**Table 4 animals-12-02081-t004:** Least squares means of rumen pH and concentrations of ammonia and volatile fatty acids (VFA).

	Treatments ^1^				
	LF ^1^	MF	HF	SEM	*p*<
Rumen pH (units)	6.32 ^b^	6.56 ^a^	6.56 ^a^	0.04	0.003
Ammonia (mg/L)	146.5	145.2	141.4	5.3	0.49
Acetate (mM)	68.8 ^a^	65.2 ^b^	66.0 ^ab^	1.22	0.04
Propionate (mM)	34.7 ^a^	32.4 ^b^	31.8 ^b^	0.86	0.05
Butyrate (mM)	17.4	16.7	16.7	0.47	0.13
Isovalerate (mM)	1.27 ^b^	1.24 ^b^	1.49 ^a^	0.09	0.04
Valerate (mM)	2.62 ^a^	2.09 ^b^	2.21 ^b^	0.11	0.001
Caproic (mM)	0.94 ^a^	0.72 ^b^	0.65 ^b^	0.07	0.03
Acetate/propionate	2.00	2.02	2.09	0.04	0.19
Total VFA ^2^ (mM)	125.8 ^a^	118.5 ^b^	118.9 ^b^	2.4	0.05

Different superscript lowercase letters within a row indicate significant difference according to the *p*-value in the last column. ^1^ Treatments: Dairy cows were fed a lactating cow ration containing (i) low fat (LF)—1.7% calcium salts of palm oil fatty acids (CS-PFA) (4.7% total fat in diet), (ii) moderate fat (MF)—2.8% CS-PFA (5.8% total fat in diet), or (iii) high fat (HF)—3.9% CS-PFA (6.8% total fat in diet). ^2^ VFA = volatile fatty acids.

**Table 5 animals-12-02081-t005:** Least squares means of total-tract apparent digestibility and digesta intake of diet components.

	Treatments ^1^				
	LF	MF	HF	SEM	*p* <
Apparent digestibility (%)					
DM	55.5 ^a^	52.8 ^ab^	50.3 ^b^	1.0	0.002
Organic matter	59.0 ^a^	58.1 ^a^	54.9 ^b^	0.91	0.005
Protein	55.7	54.5	54.6	2.03	0.32
NDF	45.1 ^a^	41.7 ^b^	36.6 ^c^	0.88	0.009
ADF	23.3 ^b^	28.6 ^a^	25.8 ^ab^	1.6	0.03
Fat	54.9 ^b^	60.2 ^ab^	65.3 ^a^	2.3	0.004
Apparent digestible intake (kg/d)					
DM	15.3 ^a^	14.7 ^ab^	13.3 ^b^	0.55	0.02
Organic matter	14.9	14.6	13.2	0.58	0.07
Protein	2.2	2.0	2.2	0.11	0.13
NDF	4.2	4.3	3.8	0.19	0.08
ADF	0.97 ^b^	1.35 ^a^	1.29 ^a^	0.09	0.02
Fat	0.72 ^c^	0.95 ^b^	1.18 ^a^	0.05	0.005

Different superscript lowercase letters within a row indicate significant difference according to the *p*-value in the last column. ^1^ Treatments: Dairy cows were fed a lactating-cow ration containing (i) low fat (LF)—1.7% calcium salts of palm oil fatty acids (CS-PFA) (4.7% total fat in diet), (ii) moderate fat (MF)—2.8% CS-PFA (5.8% total fat in diet), or (iii) high fat (HF)—3.9% CS-PFA (6.8% total fat in diet). DM = dry matter.

## Data Availability

Not applicable.

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
