# Peer review of "The Effects of High-Fat Diets from Calcium Salts of Palm Oil on Milk Yields, Rumen Environment, and Digestibility of High-Yielding Dairy Cows Fed Low-Forage Diet"

_animals, 2022, doi:10.3390/ani12162081_

Round 1
Reviewer 1 Report
This paper aims to evaluate the effects of high-fat diets from calcium salts of palm oil on milk yields, rumen environment and digestibility of high yielding dairy cows. The authors claim that calcium salts of fatty acids from palm oil is a very common supplement in dairy cows' diet, their use being justified based on the instability in grain prices led to continuing worldwide growth in the proportion of fat supplements in lactating cows' rations. However the subject seems not to be new since numerous papers (quoted in this ms) have been published on the 90s. The authors are encouraged to indicate the originality/novelty of this ms that should deserve publication in Animals.
Overall the experimental design and the analytical methods are adequate and well described. However the results and discussion are too long and should be synthesized, particularly related to the numerical description of the results and statistics and discussion with other studies.
Other comments are:
InTable 1 it is stated that values are given in g/kg of DM, but it seems that are in %; moreover the total amount if ingredients given does not sum up 100%. Mineral concentrations in the supplement (given in mg/kg) seems not correct, please review.
The Title of Table (Least squares means of milk and milk solids yields) does not fit with all the end points. Please rename or give DMI and Energy intake In Table 3
First paragraph of R&D must be deleted.
Reviewer 2 Report
Review of Manuscript Animals-1845160
The paper aimed at evaluating the effect of different levels of calcium salt of palm oil in diets of dairy cows with high proportion of concentrates. The effect of fat levels was evaluated on animal performance, ruminal fermentation parameters and total tract digestibility. I have for the authors the following comments:
Major comments
Abstract is well written and clearly reflects the content and results of the study
Introduction shortly and clearly justified the realization of this study. I would just recommend finalizing the introduction with the hypothesis of the authors
M&M were described with enough detail
Results were well and scientifically discussed and using substantial literature as support
Minor comments
L17: I think even minor negative effect are not acceptable when deciding to include or not a specific feed in cow’s diet.
L18: I am not agreeing that although all this negative effects authors still write about acceptability. It seems authors try to make all negative results harmless. This is not acceptable. Be fair with dairy farmers
L27: Are these values in % in DM basis of total diet?
L29: Give the p-value for this trend. Similarly, in L32
L43: As stated, not acceptable if you find negative effects
L61: Use here you abbreviation CS-PFA
L83: How long lasted the experiment and the main period?
L88: I assume in % in DM basis in total diet
L91: Were the diets offered as TMR? Please specify
L96: About the values of NEL. Based on which systems were these calculated? Specify and give reference in the footnote
L144: I would prefer to write instead of “total-tract digestion sample collection” “faeces sample collection”. Because the latter is what you collected: faeces
L208: But I think it is also important to remark that protein content was not affected by treatments. Additionally, MUN was increased
L223: I would like to avoid writing about this “numerical trends or differences”. Preferable make discussion based on statistical differences
L237: But in your experiment there was no effect on DMI (p = 0.20). Rewrite this sentence. Similar comment for statement in L 239-240
L248: Delete “etc”. Please never use this in a scientific paper. What should the reader understand under “etc”?
L305: Where are these results presented?
L311: It would be interesting to explain the relationship between MUN and fat in the diet. This was not covered here
L351: I think Table 4 (above) should be shifted below this statement
Round 2
Reviewer 1 Report
All my comments have been addressed by the authors